## [Peer Review File · EMBO Molecular Medicine]

Characterization and therapy of fertilization failure with HNRNPR mutations

Shiming Gan, Yangyang Li, Lin Yin, Xiaotong Yang, Chen Lou, Sisi Li, Mingde Lin, Xin Li, Wenchao Xu, Jiaming Zhou, peiran hu, Zhendong Yao, Yuan Yuan, Jian-Zhong Sheng, Chen Zhang, Wei Yang, Youjiang Li, and Hefeng Huang

Corresponding author(s): Shiming Gan (shimanggan@zju.edu.cn) , Wei Yang (yangwei@zju.edu.cn), Youjiang Li (liyoujiang26@zju.edu.cn), Hefeng Huang (huanghefg@fudan.edu.cn)

Review Timeline:

Submission Date:	12th Jun 25
Editorial Decision:	9th Jul 25
Revision Received:	10th Nov 25
Editorial Decision:	16th Dec 25
Revision Received:	21st Dec 25
Accepted:	23rd Dec 25

Editor: Zeljko Durdevic

Transaction Report:

9th Jul 2025

Dear Dr. Gan,

Thank you for the submission of your manuscript to EMBO Molecular Medicine. We have now received feedback from the three reviewers who agreed to evaluate your manuscript. As you will see from the reports pasted below, all three referees recognize interest of the study but also raise serious concerns that should be addressed in a major revision. Particular attention should be given to strengthening clinical aspect of the study as suggested by the referee #1. Additionally, performing IVF in mice to test fertilization efficiency of the SKAP2-loaded extracellular vesicles treated sperm as a potential therapeutic approach would increase the translational implication of the study. If you would like to discuss further the points raised by the referees, I am available to do so via email or video. Let me know if you are interested in this option.

Further consideration of a revision that addresses reviewers' concerns in full will entail a second round of review. EMBO Molecular Medicine encourages a single round of revision only and therefore, acceptance or rejection of the manuscript will depend on the completeness of your responses included in the next, final version of the manuscript. For this reason, and to save you from any frustrations in the end, I would strongly advise against returning an incomplete revision. Further, when submitting the revised manuscript please be sure to add institutional email addresses for Youjiang Li in the manuscript and our submission system

We would welcome the submission of a revised version within six months for further consideration. Please let us know if you require longer to complete the revision.

I look forward to receiving your revised manuscript.

Yours sincerely,

Zeljko Durdevic

Zeljko Durdevic
Senior Editor
EMBO Molecular Medicine

We require:

- 1) A .docx formatted version of the manuscript text (including legends for main figures, EV figures and tables). Please make sure that the changes are highlighted to be clearly visible.
- 2) Individual production quality figure files as .eps, .tif, .jpg (one file per figure). For guidance, download the 'Figure Guide PDF': (<https://www.embopress.org/page/journal/17574684/authorguide#figureformat>).
- 3) A .docx formatted letter INCLUDING the reviewers' reports and your detailed point-by-point responses to their comments. As part of the EMBO Press transparent editorial process, the point-by-point response is part of the Review Process File (RPF), which will be published alongside your paper.
- 4) A complete author checklist, which you can download from our author guidelines (<https://www.embopress.org/page/journal/17574684/authorguide#submissionofrevisions>). Please insert information in the checklist that is also reflected in the manuscript. The completed author checklist will also be part of the RPF.

6) It is mandatory to include a 'Data Availability' section after the Materials and Methods. Before submitting your revision, primary datasets produced in this study need to be deposited in an appropriate public database, and the accession numbers and database listed under 'Data Availability'. Please remember to provide a reviewer password if the datasets are not yet public (see <https://www.embopress.org/page/journal/17574684/authorguide#dataavailability>).

12) Author contributions: You will be asked to provide CRediT (Contributor Role Taxonomy) terms in the submission system. These replace a narrative author contribution section in the manuscript.

13) A Conflict of Interest statement should be provided in the main text.

14) Every published paper now includes a 'Synopsis' to further enhance discoverability. Synopses are displayed on the journal

webpage and are freely accessible to all readers. They include a short stand first (maximum of 300 characters, including space) as well as 2-5 one-sentences bullet points that summarizes the paper. Please write the bullet points to summarize the key NEW findings. They should be designed to be complementary to the abstract - i.e. not repeat the same text. We encourage inclusion of key acronyms and quantitative information (maximum of 30 words / bullet point). Please use the passive voice. Please attach these in a separate file or send them by email, we will incorporate them accordingly.

15) Include a Reagents and Tools Table as part of the Methods section, which can be downloaded from our author guidelines (<https://www.embopress.org/page/journal/17574684/authorguide#structuredmethods>)

**** Reviewer's comments ****

Referee #1 (Comments on Novelty/Model System for Author):

Overall, this manuscript presents a well-written and scientifically rigorous study, supported by solid experimental data and innovative approaches. The authors have produced a valuable body of work using both genetic models and extracellular vesicle-based therapeutic strategies. However, the clinical component of the study would benefit from additional detail and completion, particularly with respect to semen analysis, variant annotation, fertility testing, and IVF outcomes. It would also be highly relevant to explore PLC ζ expression in both human and mouse spermatozoa to better understand the fertilization potential. I recommend a major revision to address these points before the manuscript can be considered for publication in EMBO Molecular Medicine.

Referee #1 (Remarks for Author):

I have carefully reviewed the manuscript submitted by Gan et al., entitled "SKAP2-loaded extracellular vesicles restore HNRNPR mutation-induced spermatogenic dysfunction." The study addresses an important and novel aspect of male infertility by identifying pathogenic mutations in the HNRNPR gene and providing evidence that HNRNPR is essential for spermiogenesis. The authors convincingly demonstrate that HNRNPR plays a key role in round spermatids by regulating m6A-dependent alternative splicing. The identification of Skap2, a gene involved in F-actin assembly, as a critical downstream target of HNRNPR, provides significant mechanistic insights. Furthermore, the authors show that extracellular vesicles carrying SKAP2 can rescue sperm motility defects in both mouse models and human samples, suggesting a promising therapeutic strategy for asthenoteratozoospermia.

The study is well executed and the authors have generated a substantial amount of high-quality data. Their use of several knockout and knock-in mice, deep phenotypic characterization of these models, and extracellular vesicle therapy is commendable. However, some aspects of the clinical data remain insufficiently detailed. The definition of fertilization failure in the recruited patients should be clarified, specifying whether it was observed following multiple cycles of conventional IVF, ICSI, or both. It is also essential to provide a more comprehensive semen analysis for the patients in the main text, including quantification of motility defects, detailed morphological abnormalities, assessment of sperm vitality, and results from electron microscopy analyses. In particular, the percentage of each morphological and ultrastructural defect should be reported, with a focus on the specific flagellar sections affected.

In addition, the genetic data should be reported in the main text with greater precision. The description of the HNRNPR variants should include minor allele frequencies, results from in silico pathogenicity prediction tools, precise genomic locations, and potential impacts on splicing. Although some of this information is available in the supplementary materials, it should be presented in the main text to support the strength of the genetic evidence. The transcript reference sequence of HNRNPR used for variant annotation should also be provided. The initial size of the investigated cohort and the criteria for selecting the candidate gene panel should be clearly stated.

It is noteworthy that the immunofluorescence experiments show strong HNRNPR expression in spermatocytes. This observation should be discussed further, as it might indicate an additional functional role of HNRNPR earlier in spermatogenesis. In both human and mouse samples, the electron microscopy images reveal discontinuous and partial detachment of the acrosome, a phenotype that has also been described in spermatozoa with mutations in genes encoding perinuclear ring or theca factors, such as ACTRT1, IQCN, ACTL7A, and ACTL9. The authors are encouraged to discuss this potential overlap and investigate the expression and localization of PLC ζ in mature sperm, as defects in PLC ζ are known to impair oocyte activation and fertilization. The therapeutic assessment performed in this study is promising but remains incomplete. The authors demonstrate that treatment with SKAP2-loaded extracellular vesicles improves sperm motility, actin polymerization, and morphology. However, no fertility tests were performed using treated sperm, and conventional IVF was not applied to assess the true functional rescue. Additionally, the clinical outcomes of IVF and pregnancy in the affected patients were not reported, which limits the translational impact of the findings.

Overall, this manuscript presents a well-written and scientifically rigorous study, supported by solid experimental data and innovative approaches. The authors have produced a valuable body of work using both genetic models and extracellular vesicle-

based therapeutic strategies. However, the clinical component of the study would benefit from additional detail and completion, particularly with respect to semen analysis, variant annotation, fertility testing, and IVF outcomes. It would also be highly relevant to explore PLC ζ expression in both human and mouse spermatozoa to better understand the fertilization potential. I recommend a major revision to address these points before the manuscript can be considered for publication in EMBO Molecular Medicine.

Referee #2 (Remarks for Author):

HNRNPR functions autonomously in germ cells. It was found that HNRNPR regulates the alternative splicing of Skap2 precursor mRNA in an m6A-dependent manner, and its deficiency leads to abnormal splicing and downregulated expression of Skap2. Germ cell-specific knockout of SKAP2 mimicked the phenotypic changes caused by Hnrnpr mutations, affected F-actin cytoskeleton assembly, and resulted in abnormal acrosome development and flagellar structural defects. This study focuses on asthenoteratozoospermia in male infertility, corely exploring the role and molecular mechanism of HNRNPR in spermatogenesis, and developing potential therapeutic strategies. However, some results of this paper still need clarification.

Major points:

Regarding the EVs therapy in this paper, several points require attention. The authors selected SKAP2 as the target to rescue HNRNPR deficiency, especially in the nurturing therapy of human sperm. It is worth noting that the sperm cell changes caused by HNRNPR deficiency involve axoneme-related genes such as DRC7; abnormalities in these genes lead to MMAF and manchette abnormalities, which also result in head deformities, and this mechanism is distinct from that of SKAP2-F-actin. In addition, the effects on genes like IQCN and ACTL7A cannot be treated through the SKAP2-F-actin pathway. Particularly, these effects all occur during spermatogenesis, so there is significant controversy regarding the therapeutic intervention in mature sperm.

Minor points:

"genetic mutations" should be corrected to "genetic mutations" (Line 142).

"sertoli cells" should be capitalized as "Sertoli cells".

There is a grammatical error in the phrase "phylogenetic demonstrated" in the paragraph; the noun form "phylogenetic analysis" should be used here. The complete expression should be "Comparative analysis of amino acid sequences and phylogenetic analysis demonstrated".

In the phrase "three testis of postnatal day 35 (P35) mice", "testis" is singular and should be changed to the plural form "testes". The expression "PEGX-6P-Skap2 plasmid" is non-standard; plasmid names usually start with a lowercase "p", so it should be "pEGX-6P-Skap2 plasmid".

Grammatical and word order errors:

In the phrase "and also testis size", the word order is inappropriate and the position of "also" is incorrect. It should be adjusted to "and testis size also" to make the sentence logic more coherent.

There is an incomplete sentence structure in "(MSX2, Micro-shot listed in Supplementary Table 5."; a closing parenthesis is missing, and it should be corrected to "(MSX2, Micro-shot) listed in Supplementary Table 5."

The graphical abstract appears to be less professional.

Referee #3 (Remarks for Author):

This comprehensive study investigates the role of hnRNPR in male fertility using various mouse models and a multi-omics approach. The authors demonstrate an essential role for hnRNPR in spermiogenesis and sperm function, identifying SKAP2 as a downstream target that regulates F-actin levels. Notably, the study explores extracellular vesicle (EV)-mediated delivery of SKAP2 to restore sperm motility. The study is well-designed, but some conclusions lack sufficient support from the data. The manuscript may be considered for publication after addressing the following revisions.

Major Concerns:

1. Nuclear enrichment of hnRNPR (Extended Figure 1c): The claim of nuclear enrichment of hnRNPR from the diplotene stage to round spermatids is not supported by the images provided. Additional staining with a nuclear marker or confocal imaging is

needed to substantiate this claim.

2. Protein stability (Line 158): The claim that variants destabilize the hnRNPR protein is not fully supported without data on the turnover rate of overexpressed proteins. The authors should provide such data or temper the claim.
3. hnRNPR mRNA Levels in AcKO Mice (Extended Figure 4a): The low levels of hnRnpr mRNA in AcKO mice are unexpected, given that germ cell expression appears intact. The authors should provide an explanation for this discrepancy or additional data to clarify the observation.
4. Impaired head condensation (Figure 4c): The reported impairment in sperm head condensation is not clearly demonstrated. The authors should define the morphological criteria used for head condensation and provide clearer evidence to support this claim.
5. Cell cycle analysis (Extended Figure 6i): There appears to be a difference in the G2/M phase percentages in the knock-in (KI) group, or the reported values may contain an error. The authors should verify and clarify these results.
6. Loading efficiency of mEVs-SKAP2: The loading efficiency of SKAP2 into micro-extracellular vesicles (mEVs) is not reported. The authors should provide data on the efficiency of SKAP2 incorporation to support the feasibility of the EV-mediated delivery approach.
7. Mechanism of mEVs-SKAP2 Action (Figure 8h-j): The direct effect of mEVs-SKAP2 on spermatozoa should be independent of spermiogenesis. Thus, the observed improvement in sperm motility is unlikely to result from axoneme correction. Have the authors examined the axoneme or other ultrastructural features related to sperm motility following EV treatment? Such analysis would strengthen the mechanistic claims.
8. F-actin Polymerization (Figure 9e): The increase in protein levels does not sufficiently support the claim of enhanced F-actin polymerization. Additional experiments are needed to validate this conclusion.

Minor Concerns:

8. Nomenclature consistency: The gene/protein name is inconsistently presented as "hnRNPR" or "HNRNPR." The authors should adopt standard nomenclature and ensure consistency throughout the manuscript.
9. Figure 2b labeling: The y-axis label in Figure 2b should specify "sperm morphology" instead of "normal spermatozoa (%)" for clarity.
10. Knock-in model specificity: The manuscript does not clarify whether the knock-in model targets the RGG domain of mouse hnRNPR.
11. Image scaling (Figure 4b): The images for wild-type (WT) and KI mice in Figure 4b during the maturation phase use different scales, which hinders direct comparison. The authors should standardize the scale across images.

POINT-BY-POINT RESPONSE TO REVIEWER

Below are our specific responses to the reviewer. Please note: *Reviewer comments are presented in black font, while our responses appear in blue.* The reviewer's comments have been reproduced exactly as received, without any edits.

REVIEWER COMMENTS

Referee #1:

1. Overall, this manuscript presents a well-written and scientifically rigorous study, supported by solid experimental data and innovative approaches. The authors have produced a valuable body of work using both genetic models and extracellular vesicle-based therapeutic strategies. However, the clinical component of the study would benefit from additional detail and completion, particularly with respect to semen analysis, variant annotation, fertility testing, and IVF outcomes.

Thank you very much for your valuable suggestions. We have added additional results on semen analysis (Figure EV1A), variant annotation (Figure 1D), fertility assessment (Figure 1B), IVF (Figure 1B, 7H-K), and ICSI outcomes (Figure 1B, 2A-F).

2. It would also be highly relevant to explore PLC ζ expression in both human and mouse spermatozoa to better understand the fertilization potential. I recommend a major revision to address these points before the manuscript can be considered for publication in EMBO Molecular Medicine.

Thank you for the great suggestion. We examined the expression and localization of PLC ζ , and the results showed that, compared with wild-type mice, PLC ζ expression was decreased in both human and mouse sperm carrying the *HNRNPR* mutation (Figure 5A-F). Given previous reports that PLC ζ deficiency leads to sperm-borne oocyte activation failure following ICSI (Dai *et al*, 2022), our ICSI experiments further demonstrated that oocytes injected with *HNRNPR* mutant sperm failed to undergo activation and fertilization (Figure 1B, 2A-F). Collectively, these findings indicate that *HNRNPR* mutations impair oocyte activation by reducing PLC ζ expression.

3. I have carefully reviewed the manuscript submitted by Gan *et al.*, entitled "SKAP2-loaded extracellular vesicles restore *HNRNPR* mutation-induced

spermatogenic dysfunction." The study addresses an important and novel aspect of male infertility by identifying pathogenic mutations in the *HNRNPR* gene and providing evidence that *HNRNPR* is essential for spermiogenesis. The authors convincingly demonstrate that *HNRNPR* plays a key role in round spermatids by regulating m6A-dependent alternative splicing. The identification of Skap2, a gene involved in F-actin assembly, as a critical downstream target of *HNRNPR*, provides significant mechanistic insights. Furthermore, the authors show that extracellular vesicles carrying SKAP2 can rescue sperm motility defects in both mouse models and human samples, suggesting a promising therapeutic strategy for asthenoteratozoospermia. The study is well executed and the authors have generated a substantial amount of high-quality data. Their use of several knockout and knock-in mice, deep phenotypic characterization of these models, and extracellular vesicle therapy is commendable.

Thank you very much for your kind recognition of our work.

However, some aspects of the clinical data remain insufficiently detailed. The definition of fertilization failure in the recruited patients should be clarified, specifying whether it was observed following multiple cycles of conventional IVF, ICSI, or both.

Thank you very much for your detailed question. Among the patients recruited in our study who were unable to conceive naturally, three individuals carrying *HNRNPR* gene mutations failed to obtain transferable embryos after IVF and ICSI (Figure 1B).

4. It is also essential to provide a more comprehensive semen analysis for the patients in the main text, including quantification of motility defects, detailed morphological abnormalities, assessment of sperm vitality, and results from electron microscopy analyses.

We have now included these details (Figure EV1A-C).

5. In particular, the percentage of each morphological and ultrastructural defect should be reported, with a focus on the specific flagellar sections affected.

Thank you for your excellent suggestion. We have now provided these details (Figure EV1A-C).

6. In addition, the genetic data should be reported in the main text with greater precision. The description of the *HNRNPR* variants should include minor allele frequencies, results from in silico pathogenicity prediction tools, precise genomic

locations, and potential impacts on splicing. Although some of this information is available in the supplementary materials, it should be presented in the main text to support the strength of the genetic evidence.

Thank you very much for your constructive suggestion. We have now included these details (Figure 1D).

7. The transcript reference sequence of *HNRNPR* used for variant annotation should also be provided.

Thank you for your helpful comment. We have provided the reference transcript NM_001102398.3.

8. The initial size of the investigated cohort and the criteria for selecting the candidate gene panel should be clearly stated.

Great suggestion. We have provided a detailed explanation of the initial cohort size (Lines 155-158) and the criteria used for selecting the candidate gene panel (Lines 490-492, 501-504).

9. It is noteworthy that the immunofluorescence experiments show strong *HNRNPR* expression in spermatocytes. This observation should be discussed further, as it might indicate an additional functional role of *HNRNPR* earlier in spermatogenesis.

That's a very insightful question. We have already discussed this point (Lines 418-423).

10. In both human and mouse samples, the electron microscopy images reveal discontinuous and partial detachment of the acrosome, a phenotype that has also been described in spermatozoa with mutations in genes encoding perinuclear ring or theca factors, such as *ACTRT1*, *IQCN*, *ACTL7A*, and *ACTL9*. The authors are encouraged to discuss this potential overlap and investigate the expression and localization of PLC ζ in mature sperm, as defects in PLC ζ are known to impair oocyte activation and fertilization.

Great question. We discussed the phenotypic similarities between *HNRNPR* mutations and those caused by mutations in *ACTRT1*, *IQCN*, *ACTL7A*, and *ACTL9* (Lines 424-431). Additionally, we examined the localization and expression of PLC ζ in mature sperm from both humans and mice. Compared to normal sperm, mutant sperm exhibited disrupted PLC ζ localization and reduced expression (Figure 5A-F). As previous studies have shown, the absence of PLC ζ leads to sperm-mediated oocyte activation failure following ICSI (Zhao *et al*, 2023). In our study, we found that the

HNRNPR mutation caused oocyte activation failure (Figure 1B and 2E) and investigated whether artificial oocyte activation could restore fertility. The results showed that ICSI combined with artificial oocyte activation using Srcl₂ (AOA-Srcl₂) successfully restored fertility in both humans and mice (Figure 1B and 2E).

11. The therapeutic assessment performed in this study is promising but remains incomplete. The authors demonstrate that treatment with SKAP2-loaded extracellular vesicles improves sperm motility, actin polymerization, and morphology. However, no fertility tests were performed using treated sperm, and conventional IVF was not applied to assess the true functional rescue. Additionally, the clinical outcomes of IVF and pregnancy in the affected patients were not reported, which limits the translational impact of the findings.

Thank you very much for your valuable suggestion. We conducted in vitro fertilization (IVF) after co-incubating with EVs-SKAP2. While we observed a significant improvement in sperm motility (Figure 7A, B), there was no rescue of fertilization capacity (Figure 7H-J), as *Hnrnpr*-deficient sperm exhibited impaired PLC ζ function, which leads to oocyte activation failure. To address this, we performed IVF using a combination of SKAP2 and Srcl₂. Remarkably, this combination treatment enabled *Hnrnpr*-deficient sperm to successfully fertilize oocytes and produce offspring (Figure 7H-K), demonstrating significant functional rescue. Additionally, we reported positive pregnancy outcomes, including fertilization and high-quality embryo development, in clinical patients undergoing IVF, ICSI, and AOA (Srcl₂) (Figure 1B).

Referee #2:

1. Regarding the EVs therapy in this paper, several points require attention. The authors selected SKAP2 as the target to rescue *HNRNPR* deficiency, especially in the nurturing therapy of human sperm. It is worth noting that the sperm cell changes caused by *HNRNPR* deficiency involve axoneme-related genes such as DRC7; abnormalities in these genes lead to MMAF and manchette abnormalities, which also result in head deformities, and this mechanism is distinct from that of SKAP2-F-actin. In addition, the effects on genes like IQCN and ACTL7A cannot be treated through the SKAP2-F-actin pathway. Particularly, these effects all occur during spermatogenesis, so there is significant controversy regarding the therapeutic

intervention in mature sperm.

Thank you for your thorough review of our study. We completely agree with your observation that the mechanisms by which the absence of DRC7(Morohoshi *et al*, 2020) leads to flagellar defects, manchette abnormalities, and sperm head malformations are distinct from the SKAP2-F-actin pathway. Importantly, our study highlights the critical role of the SKAP2-F-actin axis in enhancing sperm motility via F-actin-mediated functional restoration, as demonstrated in Figure 7A, B. However, it is important to note that while SKAP2-F-actin contributes to functional restoration, it does not correct the structural defects in the flagellar axoneme (Figure EV4B-E).

To further elucidate the molecular mechanism by which SKAP2-F-actin enhances sperm motility, we measured sperm phosphate levels. Our results showed a significant increase in phosphate levels following treatment of EVs-SKAP2 (Figure 7G and EV5G), which aligns with previous studies highlighting the role of F-actin in promoting ATP hydrolysis(Kanematsu *et al*, 2022) and enhancing sperm motility(Finkelstein *et al*, 2013).

Additionally, we acknowledge that mutations or deletions in genes such as *IQCN*(Wang *et al*, 2023) and *ACTL7A*(Zhou *et al*, 2023) not only lead to abnormal sperm morphology but also impair sperm-borne oocyte activation(Dai *et al.*, 2022; Xin *et al*, 2020). As a result, improving sperm motility through SKAP2-F-actin alone is insufficient to restore fertilization capacity, as shown in Figure 7H-J. To address this limitation, we explored IVF treatment with a combination of SKAP2 and *Srcl₂*, which significantly enhanced fertilization outcomes, as shown in Figure 7H-K.

In summary, our findings suggest that SKAP2-F-actin is effective in improving sperm motility, and its combination with *Srcl₂* can overcome the failure in sperm-borne oocyte activation.

2."genetic mutations" should be corrected to "genetic mutations" (Line 142).

You're welcome! I'm glad the revision helped.

3. "sertoli cells" should be capitalized as "Sertoli cells".

Thank you for bringing this issue to our attention. We have made the necessary revisions accordingly.

4. There is a grammatical error in the phrase "phylogenetic demonstrated" in the paragraph; the noun form "phylogenetic analysis" should be used here. The complete expression should be "Comparative analysis of amino acid sequences and

phylogenetic analysis demonstrated".

Thank you for your thorough review. We have corrected the grammatical errors as suggested.

5. In the phrase "three testis of postnatal day 35 (P35) mice", "testis" is singular and should be changed to the plural form "testes".

Done

6. The expression "PEGX-6P-Skap2 plasmid" is non-standard; plasmid names usually start with a lowercase "p", so it should be "pEGX-6P-Skap2 plasmid".

Done

7. In the phrase "and also testis size", the word order is inappropriate and the position of "also" is incorrect. It should be adjusted to "and testis size also" to make the sentence logic more coherent.

Done

8. There is an incomplete sentence structure in "(MSX2, Micro-shot listed in Supplementary Table 5."; a closing parenthesis is missing, and it should be corrected to "(MSX2, Micro-shot) listed in Supplementary Table 5."

Done

9. The graphical abstract appears to be less professional.

Thank you for your valuable suggestions. We have replaced the previous cartoon-style graphical abstract with a more professional illustration.

Referee #3:

1. Nuclear enrichment of hnRNPR (Extended Figure 1c): The claim of nuclear enrichment of hnRNPR from the diplotene stage to round spermatids is not supported by the images provided. Additional staining with a nuclear marker or confocal imaging is needed to substantiate this claim.

Thank you for your insightful question. To confirm the localization and expression of hnRNPR, we utilized the nuclear localization marker SYCP3 (Synaptonemal Complex Protein 3) and re-acquired confocal images. The new images clearly show the nuclear enrichment of hnRNPR from the diplotene stage through to the round spermatid stage (Appendix Figure S1B).

2. Protein stability (Line 158): The claim that variants destabilize the hnRNPR protein is not fully supported without data on the turnover rate of overexpressed proteins. The

authors should provide such data or temper the claim.

Thank you for your suggestion. We have adjusted the statement accordingly.

3. hnRNPR mRNA Levels in AcKO Mice (Extended Figure 4a): The low levels of hnRnpr mRNA in AcKO mice are unexpected, given that germ cell expression appears intact. The authors should provide an explanation for this discrepancy or additional data to clarify the observation.

Thank you for your detailed review. The results we presented show the mRNA expression levels of *Hnrnpr* in purified Sertoli cells. To enhance clarity for readers, we have added appropriate annotations to Appendix Figure S2J.

4. Impaired head condensation (Figure 4c): The reported impairment in sperm head condensation is not clearly demonstrated. The authors should define the morphological criteria used for head condensation and provide clearer evidence to support this claim.

Thank you for your insightful question. Spermiogenesis involves both acrosome development and nuclear condensation (He *et al*, 2025). Our findings demonstrate that *Hnrnpr* mutation disrupts spermiogenesis. As a result, we have updated the term "sperm head condensation" to "spermiogenesis" in our revised text. According to established literature, spermiogenesis is defined by stages 9-16, which include elongating and elongated spermatids (Hu *et al*, 2023). Morphologically, properly condensed sperm exhibit a streamlined shape and structural integrity, with well-formed acrosomes (Fan *et al*, 2022). Our PAS staining results revealed defects in acrosome development and nuclei that lacked the streamlined appearance (Figure EV2A, B). Further electron microscopy analysis, from the Golgi to the mature phase, consistently showed loose acrosomes (indicated by red arrows) and concave nuclei (highlighted by blue asterisks), supporting the conclusion that *Hnrnpr* mutation impairs both acrosome formation and nuclear condensation during spermiogenesis (Figure EV2C-H).

5. Cell cycle analysis (Extended Figure 6i): There appears to be a difference in the G2/M phase percentages in the knock-in (KI) group, or the reported values may contain an error. The authors should verify and clarify these results.

Thank you very much for highlighting the discrepancies in our data. Based on a 100% reference, the G2/M phase should be 27.81% (Appendix Figure S5J). We have corrected the percentage accordingly.

6. Loading efficiency of mEVs-SKAP2: The loading efficiency of SKAP2 into micro-extracellular vesicles (mEVs) is not reported. The authors should provide data on the efficiency of SKAP2 incorporation to support the feasibility of the EV-mediated delivery approach.

Thank you for your valuable suggestion. In response, we have included the loading efficiency results for SKAP2 (Figure EV4A). Our findings show that the loading efficiencies of mEVs-SKAP2 were high, even more than 90%, which supports the validity of the subsequent experiments.

7. Mechanism of mEVs-SKAP2 Action (Figure 8h-j): The direct effect of mEVs-SKAP2 on spermatozoa should be independent of spermiogenesis. Thus, the observed improvement in sperm motility is unlikely to result from axoneme correction. Have the authors examined the axoneme or other ultrastructural features related to sperm motility following EV treatment? Such analysis would strengthen the mechanistic claims.

Thank you for the excellent suggestion. After co-incubating SKAP2 with sperm in vitro, we aimed to determine whether the observed improvement in motility was due to a correction of the axonemal structure. To address this, we used electron microscopy to examine the sperm axoneme following EVs-SKAP2 treatment. No significant structural differences or improvements were observed (Figure EV4B-E), suggesting that SKAP2 enhances sperm motility through the upregulation of F-ACTIN, rather than by repairing the flagellar axoneme. Additionally, previous studies have shown that F-ACTIN can promote sperm motility (Finkelstein *et al.*, 2013) and facilitate rapid ATP hydrolysis (Kanematsu *et al.*, 2022), which align with our findings in mice (Figure 7A, B, E-G) and humans (EV5A, B, D-G). Consequently, our analysis emphasizes functional recovery rather than structural improvement, supporting the therapeutic mechanism proposed in this study.

8. F-actin Polymerization (Figure 9e): The increase in protein levels does not sufficiently support the claim of enhanced F-actin polymerization. Additional experiments are needed to validate this conclusion.

Thank you for your question. To further validate F-ACTIN polymerization, we measured the F-ACTIN/G-ACTIN ratio based on the method described in a previous study (Morandell *et al.*, 2021). Our results showed that incubation with EVs-SKAP2 significantly increased the F-ACTIN/G-ACTIN ratio in both human and mouse sperm

(Figure 7E, F and EV5D-F), providing additional evidence for enhanced F-ACTIN polymerization. The F-ACTIN/G-ACTIN assay method is detailed in the Lines 720-738.

9. Nomenclature consistency: The gene/protein name is inconsistently presented as "hnRNPR" or "*HNRNPR*." The authors should adopt standard nomenclature and ensure consistency throughout the manuscript.

Thank you for your constructive suggestion. In both the manuscript and the Supplementary Information, we consistently use hnRNPR to refer to the protein, *HNRNPR* for the human gene, and *Hnrnpr* for the mouse gene.

10. Figure 2b labeling: The y-axis label in Figure 2b should specify "sperm morphology" instead of "normal spermatozoa (%)" for clarity.

Thank you for your valuable guidance. Following your suggestion, we have revised the definition in Figure EV1A. To provide a more comprehensive presentation of the semen parameters, we have integrated Figure 2b into Figure EV1A.

11. Knock-in model specificity: The manuscript does not clarify whether the knock-in model targets the RGG domain of mouse hnRNPR.

Thank you for the insightful suggestion. We have clarified that the knock-in mouse model specifically targets the RGG region of hnRNPR (Appendix Figure S2A).

12. Image scaling (Figure 4b): The images for wild-type (WT) and KI mice in Figure 4b during the maturation phase use different scales, which hinders direct comparison. The authors should standardize the scale across images.

Thank you for your suggestion. We have standardized the scale bar in the updated Figure EV2C.

References

- Dai J, Li Q, Zhou Q, Zhang S, Chen J, Wang Y, Guo J, Gu Y, Gong F, Tan Y *et al* (2022) IQCN disruption causes fertilization failure and male infertility due to manchette assembly defect. *EMBO Mol Med* 14: e16501
- Fan Y, Huang C, Chen J, Chen Y, Wang Y, Yan Z, Yu W, Wu H, Yang Y, Nie L *et al* (2022) Mutations in CCIN cause teratozoospermia and male infertility. *Sci Bull (Beijing)* 67: 2112-2123
- Finkelstein M, Megnagi B, Ickowicz D, Breitbart H (2013) Regulation of sperm motility by PIP2(4,5) and actin polymerization. *Dev Biol* 381: 62-72
- He J, Lin X, Tan C, Li Y, Su L, Lin G, Tan YQ, Tu C (2025) Molecular insights into sperm head shaping and its role in human male fertility. *Hum Reprod Update*

Hu W, Zhang R, Xu H, Li Y, Yang X, Zhou Z, Huang X, Wang Y, Ji W, Gao F *et al* (2023) CAMSAP1 role in orchestrating structure and dynamics of manchette microtubule minus-ends impacts male fertility during spermiogenesis. *Proc Natl Acad Sci U S A* 120: e2313787120

Kanematsu Y, Narita A, Oda T, Koike R, Ota M, Takano Y, Moritsugu K, Fujiwara I, Tanaka K, Komatsu H *et al* (2022) Structures and mechanisms of actin ATP hydrolysis. *Proceedings of the National Academy of Sciences* 119: e2122641119

Morandell J, Schwarz LA, Basilico B, Tasciyan S, Dimchev G, Nicolas A, Sommer C, Kreuzinger C, Dotter CP, Knaus LS *et al* (2021) Cul3 regulates cytoskeleton protein homeostasis and cell migration during a critical window of brain development. *Nat Commun* 12: 3058

Morohoshi A, Miyata H, Shimada K, Nozawa K, Matsumura T, Yanase R, Shiba K, Inaba K, Ikawa M (2020) Nexin-Dynein regulatory complex component DRC7 but not FBXL13 is required for sperm flagellum formation and male fertility in mice. *PLoS Genet* 16: e1008585

Wang Y, Chen G, Tang Z, Mei X, Lin C, Kang J, Lian J, Lu J, Liu Y, Lan F *et al* (2023) Loss-of-function mutations in IQCN cause male infertility in humans and mice owing to total fertilization failure. *Mol Hum Reprod* 29

Xin A, Qu R, Chen G, Zhang L, Chen J, Tao C, Fu J, Tang J, Ru Y, Chen Y *et al* (2020) Disruption in ACTL7A causes acrosomal ultrastructural defects in human and mouse sperm as a novel male factor inducing early embryonic arrest. *Sci Adv* 6: eaaz4796

Zhao S, Cui Y, Guo S, Liu B, Bian Y, Zhao S, Chen Z, Zhao H (2023) Novel variants in ACTL7A and PLCZ1 are associated with male infertility and total fertilization failure. *Clin Genet* 103: 603-608

Zhou X, Xi Q, Jia W, Li Z, Liu Z, Luo G, Xing C, Zhang D, Hou M, Liu H *et al* (2023) A novel homozygous mutation in ACTL7A leads to male infertility. *Molecular Genetics and Genomics* 298: 353-360

16th Dec 2025

Dear Dr. Gan,

Thank you for the submission of your revised manuscript to EMBO Molecular Medicine. I am pleased to inform you that we will be able to accept your manuscript pending the following final amendments:

1) In the main manuscript file, please do the following:

- Please address all comments suggested by our data editors listed below:

o Data availability statement

1. Please note that the specific URLs for datasets (ProteomeXchange and CNGB repositories - PXD067981 and CNP0008311) are not provided in the data availability statement.

o Figure legends:

1. Please note that the exact p values are not provided in the legends of figures 2b, f; 3c, e, g; 4i, k; 5d, f; 6g; 7a, b, f, g, j, n; EV 1a; EV 2b, d, e, f, h; EV 3b; EV 5a, b, e-g.

2. Please indicate the statistical test used for data analysis in the legends of figures 4b, c, e, g; 6b, i.

3. Please note that information related to n is missing in the legends of figures 4b, c, h; 7c, d, f, g, j, n.

4. Please note that the white asterisk is not defined in the legend of figure EV2a. This needs to be rectified.

5. Please note that blue asterisk is denoted as red asterisk in the legend of figure EV 2g. This needs to be rectified.

6. Please note that the white dashed circle is not defined in the legend of figure EV3a. This needs to be rectified.

- In Methods, provide the antibody dilutions that were used for each antibody

- Indicate in legends number and nature of replicates and exact p= values, not a range, along with the statistical test used. To keep the figures "clear" some authors found providing an Appendix table Sx with all exact p-values preferable. You are welcome to do this if you want to.

- In data availability statement leave only information about datasets deposited in public repositories. Please use the following format to report the accession number of your data:

[data type]: [full name of the resource] [accession number/identifier] ([doi or URL or identifiers.org/DATABASE:ACCESSION])

Please check "Author Guidelines" for more information.

<https://www.embopress.org/page/journal/17574684/authorguide#availabilityofpublishedmaterial>

2) Appendix: Please add page numbers to table of contents, place each legend underneath the corresponding figure and upload the file as PDF.

3) Funding: Please make sure that information about all sources of funding including all project numbers are complete in both our submission system and in the manuscript

4) Synopsis:

5) As part of the EMBO Publications transparent editorial process initiative (see our Editorial at

<http://embomolmed.embopress.org/content/2/9/329>), EMBO Molecular Medicine will publish online a Review Process File (RPF) to accompany accepted manuscripts. This file will be published in conjunction with your paper and will include the anonymous referee reports, your point-by-point response and all pertinent correspondence relating to the manuscript. Let us know whether you agree with the publication of the RPF and as here, if you want to remove or not any figures from it prior to publication.

6) Please provide a point-by-point letter INCLUDING my comments as well as the reviewer's reports and your detailed responses (as Word file).

I look forward to reading a new revised version of your manuscript as soon as possible.

Yours sincerely,

Zeljko Durdevic

Zeljko Durdevic
Senior Editor
EMBO Molecular Medicine

*** Instructions to submit your revised manuscript ***

When preparing your revised manuscript, please refer to our guidelines: <https://link.springer.com/journal/44321/submission-guidelines#cms-Revised-submissions>. We perform an initial quality control of all revised manuscripts before re-review; failure to include requested items will delay the evaluation of your revision.

We require:

- 1) A .docx formatted version of the manuscript text (including legends for main figures, EV figures and tables). Please make sure that the changes are highlighted to be clearly visible.
- 2) Individual production quality figure files as .eps, .tif, .jpg (one file per figure). For guidance, download the 'Figure Guide PDF': <https://media.springernature.com/original/springer-cms/rest/v1/content/27825798/data/v1>.
- 3) A .docx formatted letter INCLUDING the reviewers' reports and your detailed point-by-point responses to their comments. As part of the EMBO Press transparent editorial process, the point-by-point response is part of the Review Process File (RPF), which will be published alongside your paper.
- 4) A complete author checklist, which you can download from our author guidelines. Please insert information in the checklist that is also reflected in the manuscript. The completed author checklist will also be part of the RPF.
- 5) Please note that all corresponding authors are required to supply an ORCID ID for their name upon submission of a revised manuscript.
- 6) It is mandatory to include a 'Data Availability' section after the Materials and Methods. Before submitting your revision, primary datasets produced in this study need to be deposited in an appropriate public database, and the accession numbers and database listed under 'Data Availability'. Please remember to provide a reviewer password if the datasets are not yet public.

In case you have no data that requires deposition in a public database, please state so in this section. Note that the Data Availability Section is restricted to new primary data that are part of this study.
- 7) For data quantification: please specify the name of the statistical test used to generate error bars and P values, the number (n) of independent experiments (specify technical or biological replicates) underlying each data point and the test used to calculate p-values in each figure legend. The figure legends should contain a basic description of n, P and the test applied. Graphs must include a description of the bars and the error bars (s.d., s.e.m.).
- 8) At EMBO Press we ask authors to provide source data for the main manuscript figures. You will receive a separate email with instructions for providing source data with your revised manuscript, including how to upload and organize the files.
- 9) Our journal encourages inclusion of *data citations in the reference list* to directly cite datasets that were re-used and obtained from public databases. Data citations in the article text are distinct from normal bibliographical citations and should directly link to the database records from which the data can be accessed. In the main text, data citations are formatted as follows: "Data ref: Smith et al, 2001" or "Data ref: NCBI Sequence Read Archive PRJNA342805, 2017". In the Reference list, data citations must be labeled with "[DATASET]". A data reference must provide the database name, accession number/identifiers and a resolvable link to the landing page from which the data can be accessed at the end of the reference.
- 10) We replaced Supplementary Information with Expanded View (EV) Figures and Tables that are collapsible/expandable online. A maximum of 5 EV Figures can be typeset. EV Figures should be cited as 'Figure EV1, Figure EV2' etc... in the text and

their respective legends should be included in the main text after the legends of regular figures.

- the medical issue you are addressing,

- the results obtained and

- their clinical impact.

12) Author contributions: You will be asked to provide CRediT (Contributor Role Taxonomy) terms in the submission system. These replace a narrative author contribution section in the manuscript.

13) A Conflict of Interest statement should be provided in the main text.

14) Every published paper includes a 'Synopsis' to further enhance discoverability. Synopses are displayed on the journal webpage and are freely accessible to all readers. They include a short stand first (maximum of 300 characters, including space) as well as 2-5 one-sentences bullet points that summarizes the paper. Please write the bullet points to summarize the key NEW findings. They should be designed to be complementary to the abstract - i.e. not repeat the same text. We encourage inclusion of key acronyms and quantitative information (maximum of 30 words / bullet point). Please use the passive voice. Please attach these in a separate file or send them by email, we will incorporate them accordingly.

15) Include a Reagents and Tools Table as part of the Methods section, which can be downloaded from our author guidelines.

Photos 400-800 DPI

*Additional important information regarding figures and illustrations can be found at <https://media.springernature.com/original/springer-cms/rest/v1/content/27825798/data/v1>

***** Reviewer's comments *****

Referee #2 (Remarks for Author):

The authors have addressed all my concerns.

Referee #3 (Comments on Novelty/Model System for Author):

This is a comprehensive study reporting the role of hnRNPR in male fertility using various mouse models and a multi-omics approach. The authors have addressed most of my concerns in the revision.

POINT-BY-POINT RESPONSE TO EDITORIAL

Below are our specific responses to the editorial. Please note: *Editorial comments are presented in black font, while our responses appear in blue.* The editorial's comments have been reproduced exactly as received, without any edits.

Editorial comments

1. Please note that the specific URLs for datasets (ProteomeXchange and CNGB repositories - PXD067981 and CNP0008311) are not provided in the data availability statement.

Thank you for your suggestion. We have already provided the URL links (Lines 751-755).

2. Please note that the exact p values are not provided in the legends of figures 2b, f; 3c, e, g; 4i, k; 5d, f; 6g; 7a, b, f, g, j, n; EV 1a; EV 2b, d, e, f, h; EV 3b; EV 5a, b, e-g.

Great suggestion. We have provided the exact *P* value. It is worth noting that statistically, when the *P* value is less than 0.0001, we referred to the article published in *EMBO Molecular Medicine* and adopted the presentation method of $P < 0.0001$.

3. Please indicate the statistical test used for data analysis in the legends of figures 4b, c, e, g; 6b, i.

Done

4. Please note that information related to n is missing in the legends of figures 4b, c, h; 7c, d, f, g, j, n.

Done

5. Please note that the white asterisk is not defined in the legend of figure EV2a. This needs to be rectified.

Done

6. Please note that blue asterisk is denoted as red asterisk in the legend of figure EV 2g. This needs to be rectified.

Done

7. Please note that the white dashed circle is not defined in the legend of figure EV3a.

This needs to be rectified.

Done

8. In Methods, provide the antibody dilutions that were used for each antibody.

Done

9. Indicate in legends number and nature of replicates and exact p -values, not a range, along with the statistical test used. To keep the figures "clear" some authors found providing an Appendix table Sx with all exact p -values preferable. You are welcome to do this if you want to.

Thank you very much for your suggestion. We have provided the repeated numbers and properties, as well as statistical methods in the figure legends, and have marked the exact P -values in the figures. It is worth noting that when $P < 0.0001$, we referred to the recent articles from EMBO Molecular Medicine and presented it in the form of $P < 0.0001$. At the same time, since we have provided the appropriate P -values in figures, we no longer choose to provide a Appendix table Sx. Thank you sincerely for your suggestion again.

10. In data availability statement leave only information about datasets deposited in public repositories. Please use the following format to report the accession number of your data: The datasets produced in this study are available in the following databases: [data type]: [full name of the resource] [accession number/identifier] ([doi or URL or identifiers.org/DATABASE:ACCESSION])

Done

11. Appendix: Please add page numbers to table of contents, place each legend underneath the corresponding figure and upload the file as PDF.

Done

12. Funding: Please make sure that information about all sources of funding including all project numbers are complete in both our submission system and in the manuscript.

Done

13. Synopsis: - Please check your synopsis text and image before submission with your revised manuscript. Please be aware that in the proof stage minor corrections only are allowed (e.g., typos).

Done

14. As part of the EMBO Publications transparent editorial process initiative (see our Editorial at <http://embomolmed.embopress.org/content/2/9/329>), EMBO Molecular Medicine will publish online a Review Process File (RPF) to accompany accepted manuscripts. This file will be published in conjunction with your paper and will include the anonymous referee reports, your point-by-point response and all pertinent correspondence relating to the manuscript. Let us know whether you agree with the publication of the RPF and as here, if you want to remove or not any figures from it prior to publication.

Agree

15. Please provide a point-by-point letter INCLUDING my comments as well as the reviewer's reports and your detailed responses (as Word file).

Done

23rd Dec 2025

Dear Dr. Gan,

We are pleased to inform you that your manuscript is accepted for publication and is now being sent to our publisher to be included in the next available issue of EMBO Molecular Medicine.

You may qualify for financial assistance for your publication charges - either via a Springer Nature fully open access agreement or an EMBO initiative. Check your eligibility: <https://link.springer.com/journal/44321/how-to-publish-with-us>

Zeljko Durdevic
Senior Editor
EMBO Molecular Medicine

>>> Please note that it is EMBO Molecular Medicine policy for the transcript of the editorial process (containing referee reports and your response letter) to be published as an online supplement to each paper. If you do NOT want this, you will need to inform the Editorial Office via email immediately. More information is available here: <https://link.springer.com/partners/embo-press/editorial-policies#Peer%20review>